# Low-Compute Watermark Removal via Dual-Domain Natural Projection

**Pragati Shuddhodhan Meshram** [1]   **Varun Chandrasekaran** [1]

## Abstract

Effective removal of semantic watermarks requires balancing three competing objectives: *high removal success*, *low perceptual distortion*, and *low computational cost*. However, existing single-image attacks typically optimize only for the first two, achieving strong watermark suppression but relying on expensive, multi-step optimization that limits practical deployment. In this work, we show that this trade-off is fundamental: no current approach achieves all three properties simultaneously. We introduce DAWN, a lightweight, training-free attack that explicitly targets the low-cost regime while maintaining competitive removal performance. DAWN works by projecting a watermarked image onto natural-image priors in complementary frequency and semantic spaces, suppressing watermark signals that deviate from natural statistics, and then applying a decoupled perceptual-alignment step to restore visual consistency with minimal artifact. Across diverse pixel-, frequency-, and latent-space watermarking schemes, DAWN consistently reduces detectability while preserving structural and semantic fidelity, demonstrating that efficient, low-resource watermark removal is feasible with only modest perceptual degradation. Our code is available at https://github.com/Pragati-Meshram/DAWN.

## 1. Introduction

The rise of generative models such as diffusion models (Rombach et al., 2022), DALL-E (Ramesh et al., 2022) and GANs (Goodfellow et al., 2020) has transformed content creation, enabling realistic images with minimal manual

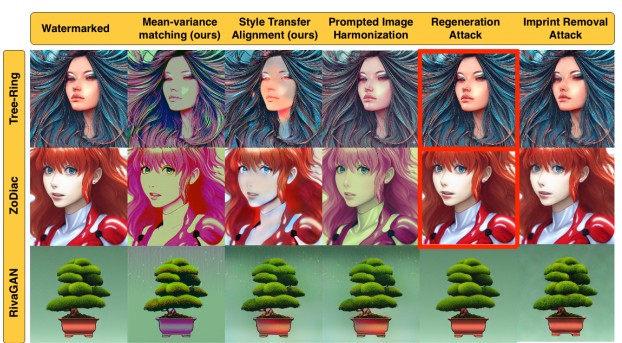

*Figure 1.* **Comparison of DAWN against existing single-image attacks across three watermarking schemes.** The *leftmost* column shows the watermarked inputs produced by each method in the corresponding row (TREE-RING (Wen et al., 2023), ZODIAC (Zhang et al., 2024), and RIVAGAN (Zhang et al., 2019)).The next three columns show outputs from DAWN with different perceptual alignment strategies (mean–variance matching, style-transfer alignment, and prompted image harmonization). The final two columns show results from prior attacks: diffusion-based regeneration (Zhao et al., 2024) and imprint-removal optimization (Müller et al., 2025). The regeneration baseline fails on semantic watermarks such as TREE-RING and ZODIAC, and the imprint-removal method requires substantially higher computational cost. Failed attack outputs (i.e., watermark still detected) are outlined with a red border.

effort. This raises concerns about provenance and authenticity, as the line between real and synthetic media blurs.

Digital watermarking has emerged as a crucial safeguard, yet adversaries increasingly target watermark forgery and removal to evade provenance tracking. Among watermarking strategies, pixel-domain methods remain the most widely deployed for their simplicity and efficiency. Early algorithms such as STEGASTAMP (Tancik et al., 2020), RIVAGAN (Zhang et al., 2019), and hybrid schemes like DWTD-CTSVD (Navas et al., 2008) embed imperceptible signatures in pixels or in frequency-domain coefficients (e.g., wavelet or discrete cosine transform coefficients). These methods withstand common distortions but are fundamentally vulnerable to regeneration attacks, where generative models recreate visually similar images without the watermark (Zhao et al., 2024). Such attacks exploit the fact that these watermarks are embedded in low-level image statistics (e.g., pixel intensities or a narrow set of DCT/wavelet coefficients) that can be modified without significantly affecting perceptual content, allowing adversaries to selectively suppress or erase the embedded signal.

[1]Department of Electrical and Computer Engineering, University of Illinois Urbana-Champaign, USA. Correspondence to: Pragati Shuddhodhan Meshram <psm12@illinois.edu>, Varun Chandrasekaran <varunc@illinois.edu>.

*Proceedings of the 43rd International Conference on Machine Learning*, Seoul, South Korea. PMLR 306, 2026. Copyright 2026 by the author(s).

To address the weaknesses of pixel- and (low dimensional) frequency-domain methods, watermarking has advanced toward more sophisticated semantic watermarking techniques that embed signals within latent or high-dimensional frequency representations during generation. Techniques such as TREE-RING (Wen et al., 2023), ZODIAC (Zhang et al., 2024), PRC-WATERMARK (Gunn et al., 2024), SFW-MARK (Lee & Cho, 2025), and FREQMARK (Guo et al., 2024) influence global image properties such as composition, texture, and structure, rather than relying on localized perturbations. By aligning signals with semantic content and leveraging frequency invariances, these approaches improve robustness against post-processing and adversarial manipulations. For instance, TREE-RING shows resilience to regeneration attacks by injecting signals directly into the frequency components of the diffusion model's latent space (Zhao et al., 2024).

Nonetheless, new vulnerabilities have emerged. Multi-image steganalysis and latent-space optimization attacks can exploit consistent artifacts across outputs or iteratively suppress watermark signals (Müller et al., 2025; Li et al., 2023). However, these attacks typically assume strong conditions such as access to multiple watermarked samples, knowledge of the underlying generator, or the computational budget for dozens to hundreds of optimization steps per image. From the perspective of semantic watermark *removal*, these methods are implicitly tuned for two of the three properties that practitioners care about most: *high removal success* and *low visual distortion*, but they ignore the third: *low compute cost*. In contrast, realistic threat models assume an adversary with only a single watermarked image, no access to the generator or watermarking scheme, and a limited compute budget, making efficiency a first-class constraint rather than an afterthought.

Building on these limitations, this paper investigates semantic watermark removal under this practical, single-image, no-box threat model. We introduce **DAWN** (**D**ual-domain **A**dversarial **W**atermark **N**ullifier), a projection-based attack designed to operate in the low-compute regime. DAWN targets the third property i.e. computational efficiency, while preserving reasonable fidelity and achieving competitive removal success across pixel-, frequency-, and semantic watermarking schemes. Our core intuition is that *watermarking, regardless of whether it is applied in pixel, frequency, or latent space, inevitably perturbs natural image statistics.* By projecting a watermarked image back toward natural priors across complementary domains, DAWN suppresses watermark artifacts using only a lightweight pipeline.

At its core, DAWN combines frequency-domain reconstruction, which attenuates structured spectral artifacts characteristic of watermark signals, with semantic refinement using pretrained generative priors to restore global coherence.

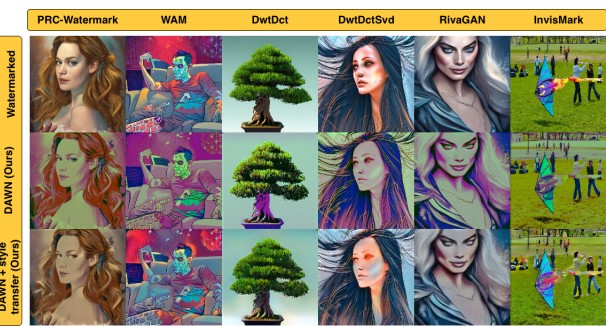

Figure 2. **DAWN removes diverse pixel-, frequency-, and latent-space watermarks while preserving the semantics of the watermarked input.** Each column shows a different watermarking method. The top row contains watermarked images; the middle row shows outputs of DAWN (introduced color hues) and the bottom row illustrates DAWN with a style-transfer alignment module that restores tone and texture without reintroducing the watermark.

Residual stylistic or chromatic deviations are orthogonal to attack success and can be addressed through lightweight perceptual-alignment modules that do not reintroduce watermark evidence. While this dual-domain suppression effectively weakens a broad range of watermark types, it does not fully guarantee perceptual fidelity across all semantic watermarking schemes, motivating a separate refinement step and revealing deeper constraints on what single-image attacks can achieve.

Across semantic watermarking schemes and three classes of single-image attacks we uncover a consistent limitation: no attack simultaneously achieves high watermark-removal success, low perceptual distortion, and low computational cost. DAWN occupies a favorable point in this landscape, balancing effectiveness and efficiency, but still cannot escape this inherent removability–distortion–compute tradeoff. Our attack samples could be seen in Figure 2. This structural limitation explains why semantic watermarking remains difficult to defeat under realistic single-image threat models.

## 2. Related Work

**Digital Watermarking Techniques:** These approaches differ along two axes: the embedding space (pixel, frequency, latent) and the embedding time i.e., whether the watermark is applied post-generation or injected in-generation during the generative process (in the initial noise or intermediate latent/frequency features). Table 1 summarizes representative methods along these two dimensions. Classical techniques such as DWTDCTSVD (Navas et al., 2008) operate in the frequency domain after image generation, whereas modern (semantic) schemes like TREE-RING (Wen et al., 2023), ZO-DIAC (Zhang et al., 2024), PRC-WATERMARK (Gunn et al., 2024), SFWMARK (Lee & Cho, 2025), FREQMARK (Guo et al., 2024), and GAUSSIAN SHADING (Yang et al., 2024b) embed watermarks in latent or frequency features during generation. Consequently, recent methods yield semantic

| Method | Pixel | Frequency | Latent | Time |
|---|---|---|---|---|
| DWTDCT (Ingemar et al., 2008) | ✗ | ✓ | ✗ | ○ |
| DWTDCTSVD (Navas et al., 2008) | ✗ | ✓ | ✗ | ○ |
| RIVAGAN (Zhang et al., 2019) | ✓ | ✗ | ✗ | ○ |
| SSL (Fernandez et al., 2022) | ✗ | ✗ | ✓ | ○ |
| ZODIAC (Zhang et al., 2024) | ✗ | ✓ | ✓ | ● |
| PRC-WATERMARK (Gunn et al., 2024) | ✗ | ✗ | ✓ | ● |
| TREE-RING (Wen et al., 2023) | ✗ | ✓ | ✓ | ● |
| WAM (Sander et al., 2024) | ✓ | ✗ | ✗ | ○ |
| INVISMARK (Xu et al., 2025) | ✓ | ✗ | ✗ | ○ |
| TRUSTMARK (Bui et al., 2023) | ✓ | ✗ | ✗ | ○ |

*Table 1.* **Categorization of watermarking methods.** ● denotes in-generation, and ○ denotes post-generation.

watermarking that align with higher-level image structure.

**Attacks on Watermarking Schemes:** The vulnerabilities of watermarking have spurred a diverse set of attack strategies. Regeneration attacks (Zhao et al., 2024) employ diffusion models to reconstruct clean images from watermarked ones, effectively erasing pixel-space signals without model access. Yet they are *far less effective* against frequency-domain schemes such as TREE-RING (Wen et al., 2023), since diffusion priors mainly suppress pixel-level perturbations while structured frequency signals often survive. Latent-space manipulation provides another avenue: Müller et al. (2025) demonstrate black-box attacks that iteratively optimize latent representations to erase or redirect watermarks. However, these approaches *depend on surrogate models and costly optimization, limiting practicality in real-time or resource-constrained settings*. Multi-image steganalysis (Yang et al., 2024a) further exploits cross-sample consistency by clustering or averaging over watermarked outputs to recover signals. However, such methods assume *access to multiple samples* per watermark key, an unrealistic condition in exemplar-free scenarios. Meanwhile, defenses emphasize the robustness of frequency-domain watermarking to standard transformations (e.g., cropping, resizing, etc.) (Wen et al., 2023). Yet these defenses overlook reconstruction-based strategies that exploit deviations from natural image statistics, leaving open pathways for low resource, single-image, model-agnostic attacks.

Existing approaches depend on surrogate models, auxiliary datasets, or costly iterative optimization. In contrast, we investigate a more practical adversarial setting: a single-image, low-resource, and model-agnostic attack that undermines both pixel- & frequency-domain watermarks by leveraging reconstruction and semantic regeneration.

## 3. Why Pixel-Only Regeneration Fails (and What We Learn)

**Hypothesis:** We hypothesize that pixel-only regeneration methods, such as Stable Diffusion img2img (Rombach et al., 2022), are insufficient for suppressing frequency-domain watermarks (e.g., TREE-RING), because they primarily regularize images in pixel space and do not explicitly

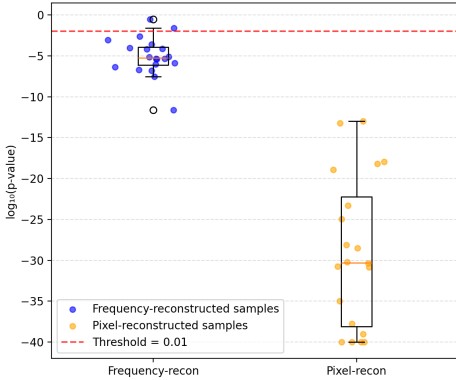

*Figure 3.* **Frequency-domain reconstruction yields much higher TREE-RING $p$-values than pixel-only diffusion, indicating stronger watermark suppression.** Blue points (Freq-recon) cluster near the 0.01 threshold, while orange points (Pixel-recon) fall far below it, showing that pixel regeneration retains watermark evidence.

constrain spectral statistics. We use img2img as a representative diffusion-based regeneration baseline, as it reflects a widely adopted and practical approach for single-image reconstruction without watermark-specific adaptation. In contrast, we posit that frequency-domain reconstruction will be more effective at weakening such watermarks by directly targeting structured spectral artifacts. We adopt a frequency-domain UNet inspired by prior work on learning natural frequency statistics (Xu et al., 2020), not as a state-of-the-art restoration method, but as a principled mechanism for enforcing natural spectral decay. We test this hypothesis by comparing diffusion-based regeneration with frequency-domain UNet reconstruction on TREE-RING watermarking.

**Approach & Setup:** We start with 20 natural images and embed TREE-RING watermarks to produce samples $\{x_w^{(i)}\}_{i=1}^{20}$. Each image is then reconstructed using two methods: (i) diffusion-based regeneration via a single pass of SD-v2 img2img (Rombach et al., 2022), and (ii) frequency-domain reconstruction by applying an $8 \times 8$ blockwise Discrete Cosine Transform (DCT), denoising the coefficients with a lightweight UNet trained on clean images, and applying the inverse DCT (IDCT). We evaluate results along two axes: (a) *watermark weakening*, measured by TREE-RING detector $p$-values ($p < 0.01$ indicates detection, $p > 0.01$ successful removal), and (b) *perceptual fidelity*, using LPIPS (Zhang et al., 2018) and CLIP similarity (Radford et al., 2021) relative to the original image.

**Results:** From Figure 3, we observe that the frequency-domain UNet (Xu et al., 2020) produced consistently higher TREE-RING $p$-values than diffusion regeneration (median $5.9 \times 10^{-6}$ vs. $5.1 \times 10^{-31}$, Wilcoxon signed-rank $p = 9.5 \times 10^{-7}$). This indicates stronger weakening of the watermark, though in most cases detection still held ($p < 0.01$). Notably, 10% of frequency-based reconstructions crossed the 0.01 threshold (successful removal), while diffusion re-

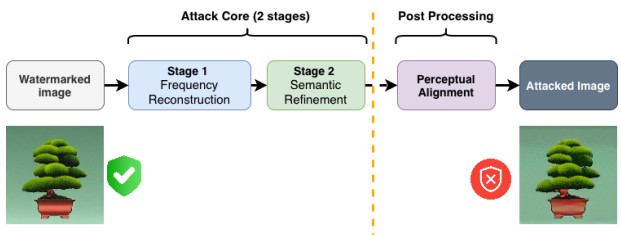

*Figure 4.* **Overview of the DAWN pipeline.** The attack core consists of two stages: (1) frequency-domain reconstruction, which suppresses structured spectral watermark artifacts, and (2) semantic refinement, which restores global coherence using a pretrained diffusion prior. A decoupled perceptual alignment step is applied as optional post-processing to correct mild hue/tone shifts but may indirectly affect detectability through chrominance realignment.

constructions universally yielded extremely low $p$-values, signifying that the watermark remained strongly detectable and the attack failed. For perceptual fidelity, however, diffusion regeneration performed better: LPIPS scores were lower (0.10 vs. 0.53) and CLIP similarity higher (0.99 vs. 0.80). Thus, diffusion models regenerate clean, semantically faithful images but fail to remove watermarks, whereas frequency reconstruction can occasionally succeed in removal but at the cost of perceptual quality.

**Takeaways:** Two guiding principles for attack design:

1. *Frequency domain projections help suppress persistent watermark signals.* They are necessary to weaken watermark traces that survive pixel-only regeneration.
2. *Semantic regeneration is helpful for preserving structure and minimizing perceptual artifacts.* It restores high-level content fidelity and reduces visual degradation introduced during watermark removal.

## 4. Our Approach: DAWN

Building on the insights from Section 3, we design DAWN (Dual-domain Adversarial Watermark Nullifier), a practical single-image attack framework that circumvents the limitations of prior methods relying on surrogate models, multi-sample access, or costly iterative optimization. The attack core of DAWN in Fig. 4 consists of two stages: (1) frequency-domain reconstruction to attenuate structured spectral anomalies associated with watermark signals, and (2) semantic refinement to restore coherent image content using pretrained generative priors. In practice, aggressive watermark suppression can introduce mild stylistic or chromatic deviations; we therefore apply a final perceptual alignment step to ensure visually consistent and usable outputs. Importantly, this alignment does not contribute to watermark suppression and is decoupled from attack success.

### 4.1. Threat Model

We consider a practical adversarial setting where the attacker has access to only a *single watermarked image* $x_w$. The goal

is to suppress or erase watermark evidence in this image so that it evades detection, while preserving semantic content and perceptual quality for downstream use.

The adversary operates under the following assumptions:

- *Knowledge.* No access to the watermarking algorithm, embedding key, detector, or generative model parameters; no auxiliary watermarked exemplars.
- *Capabilities.* The adversary may manipulate the pixels of $x_w$ using generative or restoration models, but without costly iterative optimization, surrogate watermarking models, or access to multiple samples. Moderate compute resources are assumed.
- *Scope.* The attack is image-specific rather than universal.
- *Constraints.* The attack must maintain perceptual fidelity and avoid visible artifacts.

**Cost Model.** We refer to DAWN as a *low-resource* attack because it requires *no training*, *no iterative optimization*, *no detector queries*, *no surrogate models*, and *no multi-image batches*. Its cost is dominated by a small number of feed-forward passes (inference only) through fixed pretrained networks (frequency reconstructor, diffusion refiner, optional alignment). This is substantially cheaper than optimization-based attacks whose cost grows with gradient steps or sample-dependent attacks requiring multiple inputs.

### 4.2. Detailed Approach

**Step 1: Frequency-Domain Reconstruction.** To enable watermark-agnostic suppression of spectral artifacts, we train a lightweight frequency-domain UNet $f_{\text{freq}}$ on a dataset of clean natural images $\{x_{tr}^{(i)}\}_{i=1}^{N}$. This training is performed *once*, entirely offline, and does not require watermarked samples; thus it does not contribute to the computational cost at attack time. Each image is transformed into its $8 \times 8$ blockwise DCT representation:

$$X_{tr}^{(i)} = \text{DCT}(x_{tr}^{(i)})$$

We adopt $8 \times 8$ DCT blocks because this resolution is widely used in image compression (e.g., JPEG), captures localized spectral structure, and aligns with the blockwise patterns where many frequency-domain watermarking methods embed their signals (Wen et al., 2023; Guo et al., 2024; Navas et al., 2008). These blocks provide a compact representation in which watermark perturbations often manifest as structured deviations in mid- and high-frequency bands. To mimic watermark-like distortions, we generate corruptions by injecting Gaussian noise into structured frequency bands:

$$\hat{X}^{(i)} = X_{tr}^{(i)} + \epsilon_{u,v},$$

where noise is selectively added to coefficients satisfying $t_1 \leq u + v < t_2$ within each DCT block. This simulates structured spectral deviations commonly exploited by semantic watermarking schemes such as TREE-RING and FREQMARK. To further enhance robustness, we introduce

a learnable frequency mask $M = \sigma(\theta)$ that adaptively attenuates anomalous frequency regions:

$$\hat{X}_M^{(i)} = M \cdot \hat{X}^{(i)}$$

The UNet is trained using an $\ell_1$ reconstruction objective:

$$\mathcal{L}_{\text{rec}} = \frac{1}{N} \sum_{i=1}^{N} \|f_{\text{freq}}(\hat{X}_M^{(i)}) - X_{tr}^{(i)}\|_1$$

This stage restores natural spectral decay and suppresses watermark-induced anomalies before pixel re-projection.

**Step 2: Diffusion-Based Semantic Refinement.** Although frequency reconstruction effectively weakens watermark traces, it can introduce artifacts or degrade fine textures. To restore semantic coherence, we refine the inverse-DCT output using a pretrained img2img diffusion model $\mathcal{D}$:

$$\widetilde{x}_{\text{diff}} = \mathcal{D}(\text{IDCT}(f_{\text{freq}}(\text{DCT}(x_w))))$$

This step leverages semantic priors learned by the diffusion model to recover coherent objects, textures, and global structure, while further eroding watermark signals aligned with high-level semantics.

**Optional Step 3: Perceptual Alignment for Visual Consistency.** Finally, we apply perceptual alignment to ensure visually usable outputs. In our implementation, we use lightweight channel-wise mean–variance matching:

$$\widetilde{x}_{\text{final}}^{(c)} = \sigma_c(x_w) \cdot \frac{\widetilde{x}_{\text{diff}}^{(c)} - \mu_c(\widetilde{x}_{\text{diff}})}{\sigma_c(\widetilde{x}_{\text{diff}})} + \mu_c(x_w),$$

where $\mu_c$ and $\sigma_c$ denote the mean and standard deviation of channel $c$. This step improves visual consistency with the original image but does not affect watermark detectability.

**Note:** Methods like TREE-RING modify the generative process, so the watermarked image $x_w$ is already semantically shifted relative to any unknown clean original. Our goal is therefore not to reconstruct a pre-watermarked image, but to remove the watermark (Appendix B).

### 4.3. Underneath the Hood

**Goal 1: Maximizing Attack Success.** Step 1 can be viewed as a projection of the watermarked image onto the natural-image manifold:

$$\hat{x} = \arg\min_{z \in \mathcal{M}} \|x_w - z\|_2^2,$$

where $\mathcal{M}$ denotes the natural-image manifold characterized by $1/f^\alpha$ spectral decay with $\alpha \approx 2$ for most natural images. By penalizing narrow-band spectral spikes, this projection suppresses the structured frequency energy that encodes watermark signals, directly weakening evidence that pixel-only regeneration cannot remove.

**Goal 2: Preserving Semantic Structure.** Step 2 restores global coherence, textures, and object boundaries, ensuring

that watermark suppression does not introduce structural artifacts. Perceptual alignment further improves visual consistency but is decoupled from attack success. Together, these components enable DAWN to achieve watermark suppression while maintaining output realism.

## 5. Experimental Setup

We evaluate DAWN under a strict adversarial setting where the attacker is given access to only a single watermarked image $x_w$ per trial. No auxiliary clean images, watermark keys, model parameters, or additional watermarked exemplars are available, reflecting a no-box, exemplar-free threat regime. DAWN uses a single frequency-domain UNet, which is trained *once*, offline, on clean images only. Concretely, we construct a training set of 20k images: 10k samples from the MS-COCO 2017 dataset (Lin et al., 2014) and 10k images from the Stable Diffusion Prompts (SDP) dataset, consisting of diffusion-generated outputs paired with prompts. For evaluation, we randomly select 500 clean images from each dataset, apply the target watermarking schemes to obtain their watermarked counterparts, and compute all reported metrics on these synthetically watermarked evaluation images. The training and evaluation sets are disjoint, ensuring that the frequency-domain UNet never observes watermarked images during training.

**Targeted Watermarking Methods.** We evaluate DAWN across diverse watermarking schemes spanning pixel, frequency, and latent domains, including RIVAGAN (Zhang et al., 2019), DWTDCT (Ingemar et al., 2008), DWTDCTSVD (Navas et al., 2008), SSL Watermarking (Fernandez et al., 2022), WAM (Sander et al., 2024) (32 bits), INVIS-MARK (Xu et al., 2025) (100 bits), and TRUSTMARK (Bui et al., 2023) (100 bits). More importantly, we evaluate against semantic watermarking methods TREE-RING (Wen et al., 2023), ZODIAC (Zhang et al., 2024) and PRC-WATERMARK (Gunn et al., 2024). For comparability across methods of different code lengths, we adopt detection criteria from prior work. We use $k = 32$-bit watermarks for DWTDCT, DWTDCTSVD, RIVAGAN, and SSL Watermarking. A watermark is considered detected if at least 23 out of 32 bits are correctly recovered, corresponding to a significance threshold of $p < 0.01$. This standardized setup provides a consistent basis for evaluating DAWN's effectiveness across watermarking schemes.

**Baselines and Comparative Attacks.** To contextualize DAWN's performance, we compare it against representative single-image attacks. Regeneration-based methods (Zhao et al., 2024) use Stable Diffusion img2img (Rombach et al., 2022) to project images back to the pixel manifold without watermark-specific adaptation, while imprint-removal (Müller et al., 2025) iteratively optimizes latent codes with a surrogate generator. Unlike these baselines,

| Method | PSNR ↑ | | LPIPS ↓ | | SSIM ↑ | | SSIM$_{lum}$ ↑ | | CLIP ↑ | | CLIP$_{lum}$ ↑ | | Attack Succ. ↑ | |
|---|---|---|---|---|---|---|---|---|---|---|---|---|---|---|
| | SDP | MS-COCO | SDP | MS-COCO | SDP | MS-COCO | SDP | MS-COCO | SDP | MS-COCO | SDP | MS-COCO | SDP | MS-COCO |
| TREE-RING | 14.56 | 16.12 | 0.64 | 0.67 | 0.46 | 0.45 | 0.99 | 0.99 | 0.73 | 0.69 | 0.99 | 0.99 | 70.2 | 77.4 |
| ZODIAC | 16.24 | 16.93 | 0.63 | 0.54 | 0.45 | 0.56 | 0.95 | 0.95 | 0.68 | 0.70 | 0.99 | 0.99 | 92.2 | 94.4 |
| PRC-WATERMARK | 28.23 | 28.24 | 0.56 | 0.56 | 0.55 | 0.56 | 0.99 | 0.99 | 0.77 | 0.77 | 0.99 | 0.99 | 67.8 | 65.6 |
| DWTDCT | 28.21 | 28.23 | 0.47 | 0.52 | 0.54 | 0.50 | 0.99 | 0.99 | 0.84 | 0.75 | 0.99 | 0.99 | 99.8 | 99.8 |
| DWTDCTSVD | 28.21 | 28.24 | 0.48 | 0.53 | 0.54 | 0.50 | 0.99 | 0.99 | 0.84 | 0.75 | 0.99 | 0.99 | 98.4 | 95.8 |
| RIVAGAN | 28.21 | 28.23 | 0.48 | 0.53 | 0.55 | 0.51 | 0.99 | 0.99 | 0.84 | 0.75 | 0.99 | 0.99 | 100 | 100 |
| SSL | 28.20 | 28.23 | 0.47 | 0.52 | 0.54 | 0.50 | 0.96 | 0.98 | 0.84 | 0.75 | 0.93 | 0.95 | 95.8 | 97.0 |
| WAM | 28.19 | 28.20 | 0.48 | 0.51 | 0.54 | 0.57 | 0.84 | 0.82 | 0.81 | 0.78 | 0.88 | 0.86 | 85.4 | 81.2 |
| INVISMARK | 28.20 | 28.21 | 0.47 | 0.52 | 0.56 | 0.52 | 0.91 | 0.90 | 0.82 | 0.72 | 0.89 | 0.85 | 89.0 | 90.0 |
| TRUSTMARK | 28.21 | 28.22 | 0.48 | 0.52 | 0.57 | 0.52 | 0.82 | 0.81 | 0.81 | 0.71 | 0.88 | 0.84 | 98.2 | 98.8 |

*Table 2.* **DAWN performance across watermarking methods on SDP and MS-COCO.** Attack success is driven by the dual-domain suppression core, while perceptual alignment primarily affects visual quality metrics.

DAWN requires neither surrogate models nor costly iterative optimization and still achieves higher success rates.

**Evaluation Metrics.** We evaluate along two axes: 1. For *watermark weakening*, we report detector $p$-values (e.g., for TREE-RING), where lower values indicate stronger suppression of watermark evidence (values above 0.01 correspond to successful detection); and 2. For *perceptual and semantic fidelity*, we compute PSNR, SSIM (Wang et al., 2004), and LPIPS (Zhang et al., 2018) between attacked and original images, alongside CLIP similarity (Radford et al., 2021) to capture semantic alignment. To mitigate sensitivity to color shifts, we additionally report luminance-only variants, SSIM$_{lum}$ and CLIP$_{lum}$, which isolate structural and semantic preservation in the luminance channel in YCrCb space.

**Implementation Details.** DAWN is instantiated is an inference only three-stage pipeline. First, $8 \times 8$ blockwise DCT is applied to the input image, and we train a frequency-domain UNet on mix of MS-COCO and reconstructs spectral regularities from synthetic noisy inputs using an $\ell_1$ loss (c.f. § 4). During training, we sample structured noise with standard deviations in $\{0.1, 0.2, 0.3, 0.4, 0.5, 0.6\}$ and frequency bands $(0, 5)$, $(5, 10)$, and $(10, 15)$ (DCT index ranges) to encourage robustness across low-, mid-, and high-frequency components. Second, the reconstructed image is refined using Stable Diffusion v2 img2img (Rombach et al., 2022), with frozen weights to restore semantic coherence with about 50 diffusion steps. Finally, channel-wise mean–variance matching is applied to align tone and color statistics with the original watermarked image. All learnable components are trained offline on clean data; during watermark removal, the attack itself is training-free, single-shot, and independent of the underlying generator.

**Perceptual Alignment Strategies.** DAWN incorporates a lightweight perceptual alignment step (Step 3 in the pipeline) to correct the mild tone and color shifts introduced by aggressive watermark suppression in Steps 1–2. By default, this alignment is implemented using *channel-wise mean–variance matching*, which ensures global visual consistency while remaining decoupled from watermark removal. Beyond this default choice, we further study two alternative alignment variants to assess whether different perceptual adjustments change attack behavior:

*(i) Mean–variance matching (default DAWN).* The built-in alignment module that normalizes each color channel of the attacked output to match the mean and variance of the original watermarked input.

*(ii) Style-transfer alignment.* We apply neural style transfer (Gatys et al., 2015) using the original watermarked image as the style reference. Because watermark signals are encoded in structured spectral or semantic perturbations, not global stylistic statistics. This restores chromatic and textural consistency without reintroducing watermark evidence. Examples shown in Fig. 2 and Fig. 9.

*(iii) Prompted image harmonization.* We also evaluate an external harmonization model (e.g., prompted Gemini Nano Banana (Google, 2023)), which can enhance visual realism in some cases, though with less consistency across images.

## 6. Results

Our evaluation addresses three core questions, aligned with the structure of this section:
1. **Watermark Removal Effectiveness** (§6.1): How well does DAWN suppress both classical and semantic watermarking?
2. **Perceptual and Semantic Fidelity** (§6.1): To what extent does DAWN preserve structure, semantics, and luminance?
3. **Comparison with Existing Attacks** (§6.2): How does DAWN perform relative to other attacks?

From our experiments we draw three key insights. First, DAWN reliably suppresses watermarks across both pixel- and semantic-space schemes, achieving >95% success on classical baselines and 60–90% on semantic methods (Table 2). Second, the attack preserves structure and semantics, with luminance-based metrics (SSIM$_{lum}$, CLIP$_{lum}$) consistently near 0.99 despite mild hue shifts, indicating that deviations primarily occur in color space. Finally, compared to optimization-based baselines, DAWN occupies a more favorable point in the removability–distortion–compute trade-off (§6.2): it achieves comparable or higher watermark-removal success at slightly higher distortion, while using only an inference only 50-step diffusion pass and fewer FLOPs per image than iterative imprint-removal attacks (Müller et al., 2025).

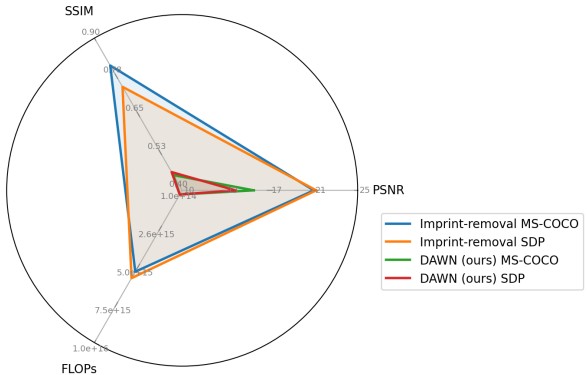

*Figure 5.* **DAWN achieves strong watermark removal at a fraction of the FLOPs required by imprint-removal.** Plot comparing FLOPs, SSIM, and PSNR on TREE-RING (SDP, MS-COCO). Imprint-removal attains higher fidelity but at far higher compute, whereas DAWN occupies the low-compute, moderate-distortion regime while maintaining competitive removal performance.

## 6.1. Effectiveness Across Watermarking Strategies

We evaluate DAWN on watermarking schemes introduced in § 5 using the SDP and MS-COCO evaluation sets. As shown in Table 2, DAWN achieves >95% success against classical methods (DWTDCT, DWTDCTSVD, RIVAGAN, SSL) while maintaining good perceptual fidelity (PSNR ~28 dB, CLIP ~0.84). For semantic schemes, success remains high (70% for TREE-RING, 90% for ZODIAC, 65% for PRC-WATERMARK), albeit with lower PSNR and higher LPIPS, reflecting the tighter coupling between watermark and latent semantics. $SSIM_{lum}$ and $CLIP_{lum}$ remain near 0.99, showing that structural and semantic information is preserved even when chromatic deviations occur. Given these, it was hard to break Gaussian-shading and SFWMark watermarks (Appendix F)

## 6.2. Comparison with Other Attacks

We compare DAWN against regeneration-based and optimization-based baselines. Distortion-based attacks (JPEG, blur, color jitter) are excluded, as prior work shows they fail to reliably remove watermarks like TREE-RING, etc. (Zhao et al., 2024; Zhang et al., 2024). As shown in Figure 5, DAWN achieves stable success in an inference only setting with 50 diffusion steps, while imprint-removal (Müller et al., 2025) requires approximately *16 × more flops* to reach comparable performance per image with approx diffusion steps of ~100. We also evaluate a regeneration baseline (Zhao et al., 2024) using Stable Diffusion v2, which again fails under our evaluation consistent with the motivating analysis in § 3, showing that pixel-only regeneration cannot suppress frequency-embedded signals such as those in TREE-RING. As shown in Figure 5, imprint-removal sits in the high-success, high-quality, *high-compute* corner of the tradeoff, requiring more FLOPs per image than DAWN occupying a high-success, low-compute,

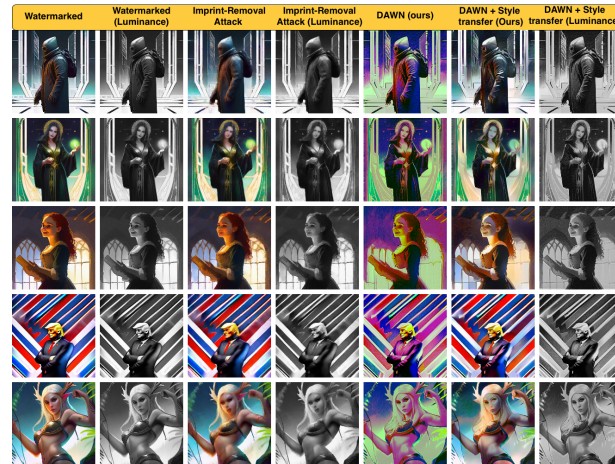

*Figure 6.* **DAWN removes TREE-RING watermarks more cleanly while preserving structure, with optional style transfer restoring color fidelity.** Comparison of watermarked inputs (left), imprint-removal outputs (Müller et al., 2025), DAWN outputs, and DAWN+STYLE TRANSFER. Every second column shows the Y-channel (luminance) in YCbCr space, confirming that DAWN maintains structural/luminance consistency even when chromatic shifts occur.

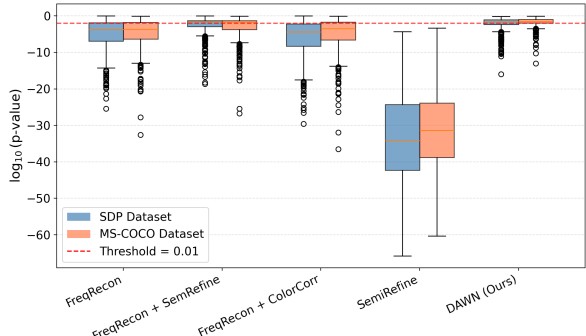

*Figure 7.* **Frequency reconstruction is the dominant driver of watermark weakening in DAWN.** Distribution of detector $p$-values ($\log_{10}$) across ablated variants on SDP and MS-COCO. Removing frequency-domain reconstruction sharply increases detector confidence (lower $p$), while the full DAWN variant consistently yields the highest $p$-values, staying closest to the 0.01 threshold.

moderate-distortion region. Regeneration lies in the remaining corner: low compute and high fidelity but near-zero success. Together, these results confirm the *removability–distortion–compute* tradeoff across single-image attacks.

Qualitatively, Figure 6 shows that while imprint-removal often leaves residual signals or semantic artifacts, DAWN suppresses watermarks more cleanly. Importantly, luminance images confirm that DAWN preserves structural fidelity, even if hue shifts emerge due to frequency perturbations.

## 6.3. Ablation and Component Analysis

To understand the role of each module in DAWN, we conduct an ablation on TREE-RING watermarks using both MS-COCO and SDP datasets. Table 3 reports metrics when

| Variant | Success↑ | | PSNR↑ | | SSIM↑ | | LPIPS↓ | | CLIP↑ | |
|---|---|---|---|---|---|---|---|---|---|---|
| | MS-COCO | SDP | MS-COCO | SDP | MS-COCO | SDP | MS-COCO | SDP | MS-COCO | SDP |
| – FreqRecon | 27.2 | 27.0 | 7.71 | 7.05 | 0.58 | 0.57 | 0.67 | 0.59 | 0.73 | 0.73 |
| – SemRefine (SD-v2) | 0.0 | 0.0 | - | - | - | - | - | - | - | - |
| – SemRefine (SDXL) | 0.0 | 0.0 | - | - | - | - | - | - | - | - |
| – FreqRecon + colorCorr | 30.0 | 23.8 | 16.99 | 17.2 | 0.67 | 0.69 | 0.57 | 0.51 | 0.77 | 0.76 |
| – FreqRecon + SemRefine (SD-v2) | 61.4 | 65.8 | 7.70 | 7.07 | 0.54 | 0.54 | 0.72 | 0.64 | 0.70 | 0.75 |
| – FreqRecon + SemRefine (SDXL) | 60.2 | 65.2 | 7.83 | 7.03 | 0.56 | 0.55 | 0.72 | 0.65 | 0.71 | 0.69 |
| **DAWN (SD-v2)** | **77.4** | **70.2** | 16.12 | 14.56 | 0.45 | 0.46 | 0.67 | 0.64 | 0.69 | 0.73 |
| **DAWN (SDXL)** | 75.2 | 68.6 | 15.68 | 15.32 | 0.51 | 0.48 | 0.68 | 0.66 | 0.71 | 0.71 |
| **DAWN (SD-v2)** + Style transfer | 71.2 | 66.4 | 18.12 | 16.66 | 0.46 | 0.48 | 0.55 | 0.47 | 0.73 | 0.79 |
| **DAWN (SDXL)** + Style transfer | 70.4 | 66.0 | 18.20 | 16.69 | 0.51 | 0.48 | 0.55 | 0.49 | 0.73 | 0.77 |

*Table 3.* **Ablation study of DAWN components on MS-COCO and SDP datasets with TREE-RING watermarks.** For modules depending on a generative backbone (SemRefine and its combinations), we report separate rows for SD-v2 and SDXL; backbone-agnostic modules (FreqRecon, FreqRecon + ColorCorr) are reported once.

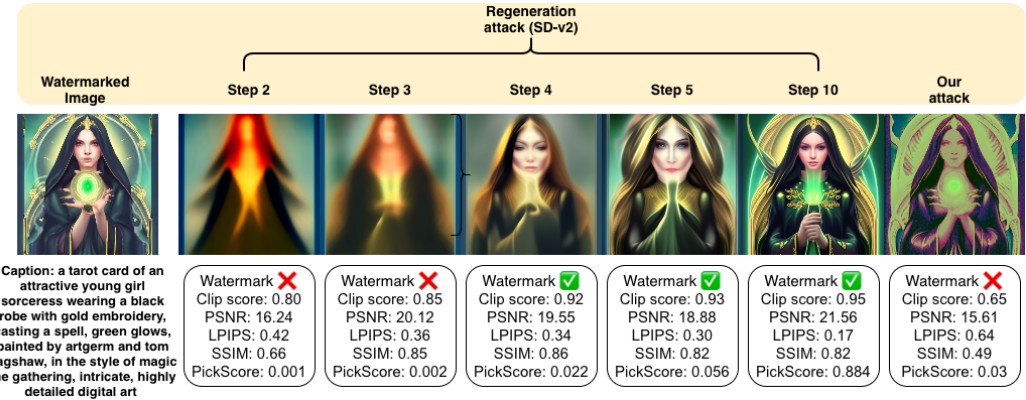

*Figure 8.* **Quality vs. watermark removal for pixel-only regeneration (SD-v2) vs. DAWN.** Small-step regeneration ($T$=2–5) yields high PSNR/LPIPS/CLIP but the watermark remains. Larger $T$ removes the watermark but causes semantic drift. DAWN achieves effective watermark weakening with better human-preference alignment (PickScore), despite lower pixel-level metrics

selectively retaining frequency-domain reconstruction (FreqRecon), semantic refinement (SemRefine), and color correction (ColorCorr). Removing *frequency-domain reconstruction* causes attack success to collapse to near zero, showing that suppressing structured spectral patterns is the dominant mechanism for watermark removal. Semantic refinement provides a more modest improvement by stabilizing high-level structure, while color correction primarily improves visual appearance by mitigating hue shifts. The full DAWN configuration and its style-transfer variant achieves the best overall balance between removal strength and perceptual quality. To quantify these differences, we compute per-image detector $p$-values for each variant (Figure 7). All DAWN configurations remain well below the 0.01 detection threshold, while removing frequency reconstruction produces a clear upward shift. This further confirms that frequency-domain projection drives watermark weakening. Style-transfer alignment improves perceptual metrics (e.g., PSNR 16.1→18.1 dB, LPIPS 0.67→0.55) while maintaining high attack success (77.4%→71.2%). Correcting hue or texture therefore does *not* resurrect the watermark signal, it trades a small success drop for substantially improved visual fidelity. Across all three perceptual-alignment variants, we observe that alignment plays only a minor role in watermark removal: mean–variance matching and style transfer yield similar suppression levels, while watermark weakening is almost entirely due to the dual-domain attack core (Steps 1–2). In contrast, prompt-driven harmonization shows inconsistent behavior (Appendix C).

### 6.4. Success ≠ Distortion Budget

Frequency perturbations can disrupt watermark signals but offer limited perceptual control, since semantics span multiple frequency bands (Appendix A). To verify that DAWN's success is not merely due to added distortion, we vary regeneration strength ($T \in \{2, 3, 4, 5, 10, 50\}$) in Fig. 8. Even at $T$=2, which introduces minimal distortion, regeneration achieves higher PSNR/SSIM/CLIP and lower LPIPS than DAWN, yet still fails to remove the watermark and often blurs important details. Its human-preference PickScore (Kirstain et al., 2023) is also substantially lower. In contrast, DAWN produces lower pixel-level metrics but consistently yields higher PickScore and much stronger watermark weakening. At comparable perceptual settings (e.g., $T$=2), DAWN removes the watermark while regeneration does not. This shows that PSNR/LPIPS/SSIM and CLIP being biased toward low-frequency similarity, do not reflect semantic fidelity in this setting (Appendix D), while PickScore does.

# 7. Conclusions

DAWN exposes fundamental weaknesses in current generative watermarking by showing that both pixel- and frequency-space watermarks can be removed with a single, training-free forward pass, revealing that many watermark designs share exploitable structural regularities in the frequency domain. These findings suggest that natural-image frequency priors can be leveraged to neutralize watermark signals without access to model parameters, posing a realistic threat to provenance mechanisms. The results highlight the need for watermarking schemes that embed signals across multiple spectral and semantic hierarchies or incorporate cryptographic authentication to remain verifiable under frequency-projection attacks.

# Impact Statement

This work studies the robustness of current watermarking schemes by analyzing how easily they can be weakened under a practical, single-image attack model. While watermark removal techniques can carry dual-use risk, our goal is to help the community understand existing vulnerabilities and design more resilient provenance mechanisms. DAWN does not enable new generative capabilities and operates only on already generated images. We encourage the use of our findings for auditing, benchmarking, and improving watermarking methods, and not for bypassing attribution or responsible model-use policies.

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

# Appendix

In this section we discuss some of the critical analysis.

## A. Limited Perceptual control.

Pixel-domain regeneration attacks admit a continuous "strength" parameter (e.g., DDIM noise level, diffusion steps), enabling smooth PSNR/LPIPS–vs–success sweeps. DAWN's frequency-domain module is fundamentally different: the watermark energy appears in different subbands across images (mid, high, or mixed-frequency rings), and our reconstruction some suppresses band frequencies. Because each image has a distinct DCT profile, there is no single scalar parameter that uniformly trades perceptual distortion against suppression strength across the dataset. Artificially introducing a global frequency-scaling sweep would not reflect how watermark energy manifests or how DAWN operates. Thus, the frequency module does not support a comparable parameter sweep.

## B. Tree-Ring watermark application.

Since, TREE-RING is an in-generation watermark our paper does *not* apply TREE-RING post-hoc to clean images. Instead, we follow the official TREE-RING implementation and *generate all watermarked images directly via Stable Diffusion with the* TREE-RING *watermark enabled*. All attacks, including DAWN and baselines, operate exclusively on these already-watermarked images. As we can see in 10, TREE-RING watermarking alters the image on embedding watermark into the images.

## C. Prompted image harmonization

We evaluate whether prompt-conditioned generative harmonization can function as a perceptual-alignment module for DAWN. For each image pair (`DAWN_attacked`$_i$, `original_watermarked_img`$_i$), we query a text-to-image model with structured prompts intended to recolor or harmonize the mid–low-frequency–refined image `DAWN_attacked`$_i$ while preserving its structure.

Our prompts span strictly constrained color-only edits to medium-strength style transfer. Examples include: (i) subtle pixel-wise hue adjustments, (ii) channel-wise histogram recoloring using `original_watermarked_img`$_i$ as a loose reference, and (iii) palette or mood transfer while keeping geometry and spatial frequencies fixed. Representative prompts instruct the model to "re-tone and recolor `DAWN_attacked`$_i$ using the soft, natural tones of `original_watermarked_img`$_i$ without altering structure" or to "blend the visual palette of `original_watermarked_img`$_i$ into

`DAWN_attacked`$_i$ at medium strength about 50% while preserving composition and objects."

Across all settings, the harmonization model shows inconsistent behavior: in some cases it fails to meaningfully adjust colors, while in others it introduces structural drift or unintended stylization. Importantly, this procedure often *reintroduces portions of the watermark signal*, reducing attack success to approx **20%**. These results indicate that prompt-driven harmonization does *not* provide a reliable alignment mechanism and can compromise both the watermark-suppressed structure and the effectiveness of DAWN, in contrast to the more stable behavior of mean–variance matching and style transfer. More prompt engineering methods can be explored in order to improve these results.

## D. Why do PSNR, LPIPS, SSIM, and CLIP all favor the $T{=}2$ regeneration image, even though it is visually less faithful?

All four metrics are biased toward low-frequency preservation and pixelwise similarity rather than semantic or perceptual alignment. At $T{=}2$, diffusion regeneration produces a heavily blurred, low-detail image that remains close to the reference in a *pixel-average* sense, which artificially inflates PSNR and SSIM and decreases LPIPS. Blurring reduces high-frequency error by definition, so PSNR and SSIM report a "better" score even though meaningful image content is lost. Similarly, CLIP similarity is dominated by global coarse structure and ignores color distortions and fine spatial detail. Thus, the $T{=}2$ image retains a high CLIP cosine despite exhibiting melted textures, loss of facial features, and poor human-perceived fidelity.

In contrast, DAWN reconstructs and restores high-frequency structure through its frequency-domain module and refinement step. These semantically meaningful details are not rewarded by PSNR/SSIM, which treat high-frequency corrections as "errors," nor fully captured by CLIP. However, they are recognized by preference-aligned metrics such as PickScore, which evaluates human-perceived semantic fidelity, composition, and aesthetic coherence. DAWN therefore scores lower on pixel-level metrics but higher on human-aligned quality criteria while achieving substantially stronger watermark suppression.

## E. Distortion attacks as baselines

Table 4 reports full results across all 10 schemes and 6 distortion types, directly addressing this concern. Results empirically confirm the **removability-distortion tradeoff**: (i) high-success distortions (Rotation, Crop) achieve removal only at PSNR $< 15$ dB — images are visually unusable; (ii) quality-preserving distortions (JPEG, Blur) maintain high PSNR but fail entirely on semantic watermarks

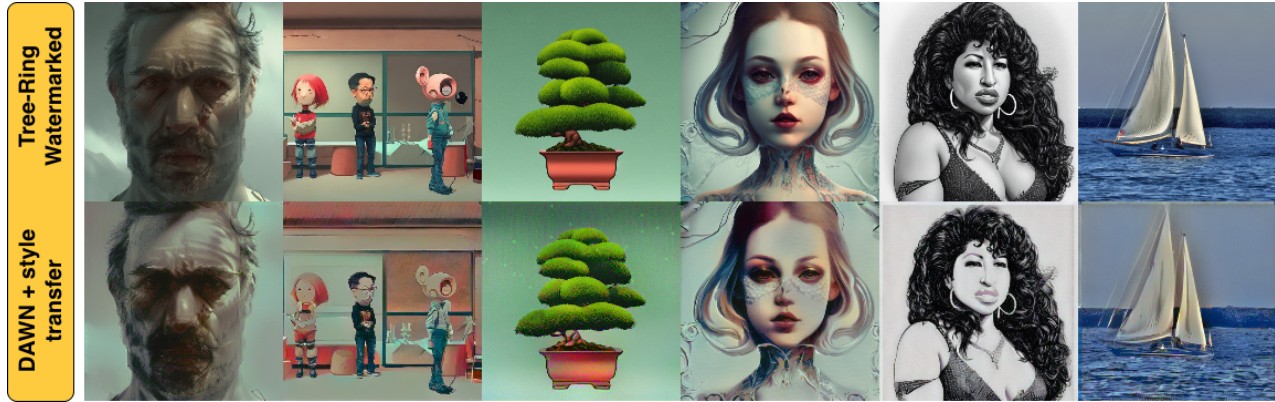

*Figure 9*. **DAWN + Style Transfer produces visually aligned outputs while preserving watermark suppression.** Examples of Tree-Ring–watermarked images (top row) and their corresponding DAWN + style-transfer results (bottom row)

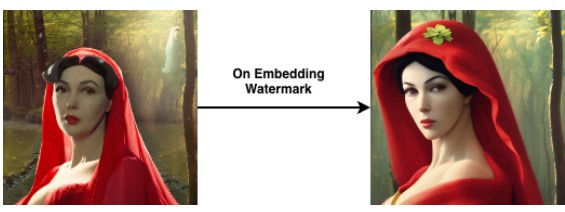

*Figure 10*. **Tree-Ring alters the generative trajectory, producing a semantically shifted watermarked image.** Comparison of a clean Stable Diffusion output (left) and its Tree-Ring–watermarked counterpart (right), generated using the official implementation. This illustrates that semantic watermarking like TREE-RING inherently modifies the final image relative to the clean baseline

(0% on Tree-Ring, 7% on Zodiac). No single distortion achieves high success *and* acceptable quality simultaneously. DAWN uniquely occupies this favorable region (74% success / PSNR 14.6 dB on Tree-Ring; >95% / PSNR ∼28 dB on classical methods), providing a qualitatively different capability that no simple distortion can replicate.

## F. Why DAWN Fails on Gaussian Shading and SFWMark

Our attack consistently fails on *Gaussian Shading* and *SFWMark* because these watermarking schemes are built around *distribution-preserving latent manipulations*, rather than the narrow-band or structured spectral artifacts that DAWN is designed to suppress. Gaussian Shading embeds watermark bits by mapping them into interval-restricted samples of the latent Gaussian distribution, ensuring that the modified latent $z_T^s$ remains statistically indistinguishable from an unwatermarked latent. As stated in the paper:

> "$z_T^s$ follows the same distribution as the randomly sampled latent representation $z_T \sim \mathcal{N}(0, I)$."

Because DAWN projects watermarked images toward natural frequency priors, it is effective only when the water-

mark introduces *detectable spectral biases*—such as the structured annular frequency energy in TREE-RING. Gaussian Shading explicitly avoids such biases, meaning that no structured frequency or semantic perturbation exists for our frequency-domain module to remove.

Similarly, *SFWMark* encodes watermark signals through semantic flow warping and spatially varying signal injection, neither of which manifests as a stable spectral residual or a correctable chromatic shift. Its watermark is coupled to spatial deformation fields and viewpoint-consistent texture perturbations, so DAWN's frequency projection suppresses neither the warped feature trajectories nor the learned semantic perturbations that survive reconstruction.

Both watermarking method therefore conceal their signals in aspects of the latent representation or spatial geometry that remain intact after DAWN's frequency reconstruction and semantic refinement. These findings highlight a fundamental limitation of frequency-based single-image watermark removal and suggest that future attacks may require geometric or latent-manifold manipulations rather than purely spectral projections.

| Method | Distortion Attack | PSNR ↑ | LPIPS ↓ | SSIM ↑ | SSIM$_{lum}$ ↑ | CLIP ↑ | CLIP$_{lum}$ ↑ | Attack Succ. (%) ↑ |
|---|---|---|---|---|---|---|---|---|
| TREE-RING | Rotation (75°) | 9.68 | 0.70 | 0.35 | 0.36 | 0.86 | 0.83 | 50 |
| | JPEG (Q=25) | - | - | - | - | - | - | 0 |
| | Crop (0.75) | 13.43 | 0.45 | 0.50 | 0.51 | 0.96 | 0.96 | 27 |
| | Blur ($\sigma$=8) | - | - | - | - | - | - | 0 |
| | Noise ($\sigma$=0.1) | 12.68 | 0.97 | 0.16 | 0.22 | 0.84 | 0.86 | 35 |
| | Brightness (1.5×) | 4.74 | 0.72 | 0.47 | 0.49 | 0.72 | 0.76 | 6 |
| ZODIAC | Rotation (75°) | 10.86 | 0.60 | 0.33 | 0.33 | 0.79 | 0.75 | 89 |
| | JPEG (Q=25) | 31.66 | 0.07 | 0.88 | 0.91 | 0.92 | 0.95 | 24 |
| | Crop (0.75) | 12.93 | 0.47 | 0.38 | 0.39 | 0.92 | 0.92 | 90 |
| | Blur ($\sigma$=8) | 21.19 | 0.62 | 0.59 | 0.60 | 0.75 | 0.70 | 72 |
| | Noise ($\sigma$=0.1) | 20.8 | 0.50 | 0.29 | 0.42 | 0.91 | 0.89 | 7 |
| | Brightness (1.5×) | 23.83 | 0.04 | 0.91 | 0.92 | 0.96 | 0.97 | 4 |
| PRC-WATERMARK | Rotation (75°) | 28.00 | 0.70 | 0.32 | 0.98 | 0.86 | 0.99 | 100 |
| | JPEG (Q=25) | 34.57 | 0.17 | 0.86 | 0.99 | 0.91 | 0.99 | 6 |
| | Crop (0.75) | 28.30 | 0.55 | 0.42 | 0.99 | 0.94 | 0.99 | 100 |
| | Blur ($\sigma$=8) | 29.98 | 0.76 | 0.64 | 0.99 | 0.76 | 0. | 100 |
| | Noise ($\sigma$=0.1) | 28.40 | 0.77 | 0.17 | 0.36 | 0.94 | 0.99 | 18 |
| | Brightness (1.5×) | - | - | - | - | - | - | 0 |
| DWTDCT | Rotation (75°) | - | - | - | - | - | - | 0 |
| | JPEG (Q=25) | 25.49 | 0.06 | 0.91 | 0.99 | 0.97 | 0.98 | 61 |
| | Crop (0.75) | - | - | - | - | - | - | 0 |
| | Blur ($\sigma$=8) | - | - | - | - | - | - | 0 |
| | Noise ($\sigma$=0.1) | 23.23 | 0.31 | 0.54 | 0.97 | 0.82 | 0.98 | 72 |
| | Brightness (1.5×) | 25.49 | 0.06 | 0.91 | 0.98 | 0.87 | 0.99 | 61 |
| DWTDCTSVD | Rotation (75°) | - | - | - | - | - | - | 0 |
| | JPEG (Q=25) | 28.80 | 0.01 | 0.77 | 0.98 | 0.94 | 0.98 | 5 |
| | Crop (0.75) | - | - | - | - | - | - | 0 |
| | Blur ($\sigma$=8) | 21.75 | 0.69 | 0.68 | 0.99 | 0.75 | 0.74 | 2 |
| | Noise ($\sigma$=0.1) | 25.23 | 0.36 | 0.99 | 0.99 | 0.99 | 0.99 | 100 |
| | Brightness (1.5×) | 25.49 | 0.06 | 0.91 | 0.98 | 0.87 | 0.99 | 61 |
| RivaGAN | Rotation (75°) | - | - | - | - | - | - | 0 |
| | JPEG (Q=25) | 29.80 | 0.01 | 0.81 | 0.99 | 0.94 | 0.97 | 77 |
| | Crop (0.75) | 19.21 | 0.42 | 0.43 | 0.97 | 0.94 | 0.99 | 98 |
| | Blur ($\sigma$=8) | - | - | - | - | - | - | 0 |
| | Noise ($\sigma$=0.1) | 25.21 | 0.03 | 0.99 | 0.99 | 0.99 | 0.99 | 72 |
| | Brightness (1.5×) | 25.49 | 0.06 | 0.91 | 0.98 | 0.97 | 0.99 | 61 |
| SSL | Rotation (75°) | 27.97 | 0.70 | 0.37 | 0.39 | 0.85 | 0.82 | 22 |
| | JPEG (Q=25) | 30.86 | 0.12 | 0.81 | 0.85 | 0.94 | 0.97 | 92 |
| | Crop (0.75) | 28.31 | 0.55 | 0.43 | 0.45 | 0.94 | 0.94 | 27 |
| | Blur ($\sigma$=8) | 29.56 | 0.67 | 0.64 | 0.65 | 0.77 | 0.76 | 99 |
| | Noise ($\sigma$=0.1) | 28.57 | 0.51 | 0.26 | 0.39 | 0.96 | 0.94 | 98 |
| | Brightness (1.5×) | 30.8 | 0.08 | 0.91 | 0.94 | 0.97 | 0.98 | 13 |
| INVISMARK | Rotation (75°) | 9.91 | 0.70 | 0.40 | 0.41 | 0.85 | 0.83 | 92 |
| | JPEG (Q=25) | 30.11 | 0.13 | 0.83 | 0.86 | 0.93 | 0.97 | 66 |
| | Crop (0.75) | - | - | - | - | - | - | 0 |
| | Blur ($\sigma$=8) | 20.82 | 0.67 | 0.66 | 0.67 | 0.77 | 0.76 | 99 |
| | Noise ($\sigma$=0.1) | 20.89 | 0.60 | 0.25 | 0.38 | 0.96 | 0.93 | 13 |
| | Brightness (1.5×) | - | - | - | - | - | - | 0 |
| TRUSTMARK | Rotation (75°) | 27.97 | 0.70 | 0.39 | 0.41 | 0.85 | 0.83 | 100 |
| | JPEG (Q=25) | 32.78 | 0.12 | 0.83 | 0.86 | 0.93 | 0.97 | 99 |
| | Crop (0.75) | 28.31 | 0.56 | 0.45 | 0.47 | 0.94 | 0.94 | 83 |
| | Blur ($\sigma$=8) | 29.30 | 0.66 | 0.64 | 0.66 | 0.77 | 0.76 | 53 |
| | Noise ($\sigma$=0.1) | 28.56 | 0.53 | 0.27 | 0.41 | 0.95 | 0.93 | 100 |
| | Brightness (1.5×) | 30.06 | 0.09 | 0.90 | 0.93 | 0.97 | 0.98 | 50 |
| WAM | Rotation (75°) | 27.97 | 0.71 | 0.39 | 0.41 | 0.86 | 0.83 | 16 |
| | JPEG (Q=25) | - | - | - | - | - | - | 0 |
| | Crop (0.75) | - | - | - | - | - | - | 0 |
| | Blur ($\sigma$=8) | 29.58 | 0.67 | 0.66 | 0.68 | 0.77 | 0.76 | 9 |
| | Noise ($\sigma$=0.1) | - | - | - | - | - | - | 0 |
| | Brightness (1.5×) | - | - | - | - | - | - | 0 |

*Table 4.* Robustness under common image distortions including geometric, compression, noise, and photometric transformations. Entries with 0% attack success show '-' for quality metrics as the attacked image is indistinguishable from the original under detection. High-success distortions (Rotation, Crop) incur severe quality loss (PSNR < 15 dB), while quality-preserving distortions (JPEG, Blur) fail on semantic watermarks, confirming the removability–distortion tradeoff.

