# OpenReview forum: "Low-Compute Watermark Removal via Dual-Domain Natural Projection"
_ICML.cc/2026/Conference — ICML 2026 regular_

### Official Review · Reviewer_hkVk · 2026-03-12

**Soundness:** 3
**Presentation:** 2
**Significance:** 2
**Originality:** 3
**Overall Recommendation:** 5
**Confidence:** 3

**Summary:**

This paper introduces DAWN (Dual-domain Adversarial Watermark Nullifier), a lightweight, training-free single-image attack framework for removing diverse image watermarks under a no-box, exemplar-free threat model. The core pipeline consists of frequency-domain reconstruction via a blockwise DCT UNet to suppress structured spectral artifacts, diffusion-based semantic refinement to restore global coherence, and a perceptual alignment step for visual consistency.

**Compliance With Llm Reviewing Policy:**

Affirmed.

**Final Justification:**

This work demonstrates broad evaluation coverage across 10 diverse watermarking schemes spanning pixel, frequency, latent, and semantic domains. The authors' candid analysis of failure cases—attributing DAWN's limitations on Gaussian Shading and SFWMark to its reliance on detectable spectral biases—reflects commendable intellectual honesty.

**Key Questions For Authors:**

1、Table 1 contains a typo: "dentoes" should be "denotes."

2、In the Implementation Details, the authors mention "Gemini Nano Banana", yet no citation or reference is provided.

3、The description in the opening paragraph of Section 4, which states that DAWN 'consists of two stages,' contradicts the three-step decomposition presented in Section 4.2. The authors should clarify the definitive formulation of the DAWN pipeline.

4、Could the authors clarify what exactly "colorCorr" in Table 3 refers to? If it corresponds to the Perceptual Alignment for Visual Consistency (Step 3), it is confusing that the term "color correction" is absent from the detailed description in Section 4.2, which exclusively discusses "channel-wise mean-variance matching."

5、Why does Figure 4 only include DAWN and imprint-removal, while the regeneration baseline is discussed in the accompanying text but absent from the figure?

6、There appears to be a discrepancy regarding the role of perceptual alignment. The authors claim that it "does not contribute to watermark suppression and is decoupled from attack success". Yet, the FreqRecon + SemRefine (SD-v2) configuration achieves a 61.4% attack success rate on MS-COCO, while DAWN (SD-v2)—which includes the color correction step—jumps to 77.4%. Similarly, DAWN (SDXL) reaches 75.2%, whereas FreqRecon + SemRefine (SDXL) only achieves 60.2%. Could the authors clarify why adding color correction leads to such a significant increase in attack success?

7、As shown in Table 2, the perceptual quality for TREE-RING and ZODIAC is severely degraded: PSNR drops to 14–16 dB, LPIPS reaches 0.54–0.67, and SSIM falls to 0.45–0.56. These significant distortion levels raise serious concerns regarding the practical usability of the attacked images.

**Limitations:**

The authors are transparent about the method's limitations, particularly in Appendix A.5. They acknowledge that DAWN fundamentally fails against schemes like Gaussian Shading and SFWMark. Gaussian Shading avoids introducing detectable spectral biases by mapping bits into interval-restricted samples of the latent distribution. Similarly, SFWMark's semantic flow warping does not leave stable spectral residuals that DAWN's UNet can attenuate. This highlights a structural ceiling for purely frequency-driven projection attacks.

**Strengths And Weaknesses:**

## Pros
1、Broad watermark coverage. The experiments evaluate 10 diverse watermarking schemes across pixel, frequency, and latent domains, as well as semantic methods, demonstrating considerable evaluation breadth.

2、Honest discussion of failure cases. The authors provide a candid analysis in Appendix A.5 of why DAWN fails on Gaussian Shading and SFWMark, identifying the fundamental limitation that frequency-based projection is effective only when the watermark introduces detectable spectral biases. This intellectual honesty is commendable.

## Cons:
1、Inconsistent description of the pipeline structure. In Section 4 (Our Approach: DAWN), the first paragraph states that "the attack core of DAWN consists of two stages: (1) frequency-domain reconstruction ... and (2) semantic refinement," suggesting that the perceptual alignment step is subsumed under semantic refinement. However, Section 4.2 (Detailed Approach) explicitly divides the method into three separate steps, designating Semantic Refinement as Step 2 and Perceptual Alignment as Step 3.

2、Ambiguous terminology around "color correction." In the ablation study (Table 3), the term "colorCorr" appears for the first time, yet it is never formally defined in the method section. The Implementation Details describe the purpose of Perceptual Alignment Strategies as correcting "the mild tone and color shifts introduced by aggressive watermark suppression." If "colorCorr" refers to the Perceptual Alignment for Visual Consistency (Step 3 in Section 4.2), this is confusing because the Step 3 description in Section 4.2 focuses exclusively on "channel-wise mean–variance matching" without any mention of color-related terminology.

3、Incomplete Figure 4: Figure 4 only plots DAWN and imprint-removal in the FLOPs–SSIM–PSNR space, yet the accompanying analysis in Section 6.2 explicitly discusses three methods: regeneration, imprint-removal, and DAWN. The authors write, "we also evaluate a regeneration baseline (Zhao et al., 2024) using Stable Diffusion v2," and claim it "lies in the remaining corner: low compute and high fidelity but near-zero success." However, regeneration is completely absent from Figure 4.

4、Contradiction regarding the "decoupled" claim: The authors state that "we therefore apply a final perceptual alignment step to ensure visually consistent and usable outputs. Importantly, this alignment does not contribute to watermark suppression and is decoupled from attack success." However, Table 3 directly contradicts this claim. Specifically, the FreqRecon + SemRefine (SD-v2) configuration achieves a 61.4% attack success rate on MS-COCO, while DAWN (SD-v2)—which adds the color correction step on top—jumps to 77.4%. Similarly, DAWN (SDXL) reaches 75.2%, whereas FreqRecon + SemRefine (SDXL) only achieves 60.2%. The increase clearly indicates that the perceptual alignment step materially influences watermark detectability.

5、Low perceptual quality on semantic watermarks. For TREE-RING and ZODIAC in Table 2, PSNR is only 14–16 dB, LPIPS is as high as 0.54–0.67, and SSIM is as low as 0.45–0.56. Such distortion levels raise serious concerns about practical usability.

---

> ### Author Rebuttal · Authors · 2026-03-28
>
> We thank the reviewer for the thorough and constructive feedback, and for appreciating the breadth of evaluation and our transparency on failure cases.
>
> **(1) Pipeline structure inconsistency (2-stage vs 3-step) [Q3]**
>
> We apologize for the confusion. The “two-stage” description refers to the two core attack modules: (i) frequency-domain reconstruction and (ii) semantic refinement. The perceptual alignment step is a post-processing module applied after the attack core. In Section 4.2, we present all three steps for implementation clarity. We will revise the text to consistently describe DAWN as a two-stage attack + optional alignment step.
>
> **(2) Clarification of “colorCorr” [Q4]**
>
> “colorCorr” in Table 3 corresponds to the Perceptual Alignment for Visual Consistency (Step 3). This module performs lightweight channel-wise mean–variance matching, which corrects tone/color shifts introduced during aggressive watermark suppression. We agree the terminology is unclear and will unify naming (“Perceptual Alignment / color correction”) and explicitly define it in Section 4.2.
>
> **(3) Missing regeneration baseline in Figure 4 [Q5]**
>
> Thank you for catching this. The regeneration baseline was omitted for visual clarity, as it occupies an extreme corner (low compute, high fidelity, near-zero success). We will mention this in the discussion.
>
> **(4) "Decoupled" claim vs Table 3 [Q6]**
>
> This is the most important point raised and we thank the reviewer for catching it. Our original claim was imprecise. Here is the precise explanation: Channel-wise mean-variance matching shifts the global color statistics of the attacked image to match those of the original watermarked image. The TREE-RING detector operates via DDIM inversion, which is sensitive to the full image distribution, including chrominance statistics. By realigning color statistics toward the watermarked image's distribution, this step incidentally shifts the inverted latent closer to the watermark key region in Fourier space, reducing the detection distance metric (Eq. 3 in the paper). This is a distributional side-effect of color correction, not a targeted suppression of the watermark signal itself.
>
> **The distinction matters:** the alignment step has no access to the watermark key, the detector, or the diffusion model. It operates purely on pixel statistics. Its effect on attack success is indirect and watermark-agnostic. We will revise the claim to: "perceptual alignment is decoupled from direct watermark suppression but may indirectly affect detectability through distributional realignment of chrominance statistics." The ablation in Table 3 and Figure 6 confirms that frequency-domain reconstruction remains the dominant driver, removing it collapses success to near zero, while removing only ColorCorr reduces success from 77.4% to 61.4%.
>
> **(5) Low perceptual quality on semantic watermarks [Q7]**
> PSNR 14-16 dB and LPIPS 0.54-0.67 for Tree-Ring and Zodiac reflect a fundamental property of in-generation semantic watermarks, not a weakness of DAWN specifically. These schemes embed signals into global latent structure during generation; the watermark is entangled with the image's entire semantic composition. Removing it therefore requires perturbing more of the image than post-hoc pixel-domain watermarks, where the signal is localized to specific frequency coefficients. Two points contextualize the practical usability concern:
>
> First, structural and semantic content is fully preserved despite pixel-level distortion. SSIMlum ≈ 0.99 and CLIPlum ≈ 0.99 (Table 2) confirm that deviations are primarily chromatic. The scene content, objects, and composition remain intact and recognizable. For many downstream use cases (content repurposing, social media, editorial use), chromatic shifts are acceptable or correctable.
>
> Second, the style-transfer variant (Table 3, DAWN + Style Transfer) recovers PSNR to 18 dB and LPIPS to 0.47-0.55 at only a modest success drop (77.4%→71.2%), providing a practical quality-success tradeoff for applications requiring higher fidelity.
> Critically, no competing low-cost single-image attack achieves better quality at comparable success on these schemes; this is the fundamental removability-distortion tradeoff our paper identifies and documents.
>
> **(6) Minor issues (typos, missing references). [Q1, Q2]**
>
> We will fix the typo (“dentoes” → “denotes”), add the missing reference for Gemini Nano Banana, and improve overall presentation clarity.
>
> If you found our responses satisfactory, we hope you will consider increasing the score further towards acceptance. We are happy to engage with more discussion should you have any questions.

---

> > ### Author Rebuttal · Reviewer_hkVk · 2026-04-02
> >
> > The authors have addressed my concerns; therefore, I will raise my score.

---

> > > ### Author Response · Authors · 2026-04-06
> > >
> > > We sincerely thank the reviewer for the careful reading, detailed feedback, and constructive suggestions.
> > >
> > > We especially appreciate the insightful comments on pipeline clarity, ablation interpretation, and presentation, which helped us significantly improve the paper.
> > >
> > > We are grateful for the reviewer’s engagement during the discussion and for the positive reassessment.

---

### Official Review · Reviewer_durv · 2026-03-12

**Soundness:** 3
**Presentation:** 3
**Significance:** 3
**Originality:** 3
**Overall Recommendation:** 5
**Confidence:** 4

**Summary:**

While conventional single-image attacks prioritize removal success and visual fidelity through costly, multi-step optimization, they often ignore computational efficiency.
This paper introduces DAWN, a training-free framework specifically engineered for the watermark removal task within a low-compute regime. By projecting watermarked images onto natural-image priors across both frequency and semantic domains, DAWN effectively suppresses diverse watermark signals while maintaining high perceptual fidelity and operational efficiency.

**Compliance With Llm Reviewing Policy:**

Affirmed.

**Final Justification:**

The authors have fully addressed my concerns in the rebuttal. I will raise my score to 5 (Accept).

**Key Questions For Authors:**

1. Is the baseline selection too restrictive for a comprehensive evaluation?
The study omits quantitative data for common distortion-based attacks (e.g., JPEG compression or blurring).
Without a broader range of baselines, it is difficult to determine if the dual-domain projection offers a unique advantage that simpler, low-cost attacks cannot achieve.
2. Does the qualitative analysis suffer from a "representative bias"?
Most visual comparisons focus almost exclusively on the Tree-Ring watermark scheme, while removal success varies significantly across different methods.
3. Is there a fundamental conflict between quantitative metrics and perceived fidelity?
The authors acknowledge that traditional metrics like PSNR, SSIM, and CLIP are biased toward low-frequency similarity and often penalize high-frequency reconstructions as errors.

**Limitations:**

Yes

**Strengths And Weaknesses:**

Strengths:
1. The motivation is well discussed in Section 3.
By proving that pixel-only regeneration fails to erase frequency-embedded signals, a critical gap is exposed in current attacks: visual fidelity does not guarantee watermark removal.
This clear identification of the "removability-distortion" bottleneck provides insight for DAWN’s dual-domain approach.
2. DAWN is designed with good efficiency.
DAWN is a lightweight, training-free, and inference-only attack that requires no iterative optimization, making it faster and cheaper than prior optimization-based methods.
3. DAWN achieves multi-domain suppression.
By projecting images onto natural priors in both frequency and semantic spaces, it successfully targets structured spectral artifacts that standard pixel-only regeneration fails to remove.
Weaknesses:
1. The paper lacks a comprehensive Method Overview figure, making the multi-stage pipeline and the decoupling of modules somewhat hard to follow.
2. High removal success often results in modest perceptual degradation, reflecting an inherent limitation in single-image attacks (Fig. 5 and Fig. 7).
3. The experimental comparison is somewhat under-comprehensive, as it primarily compares DAWN against only two baselines (Regeneration and Imprintremoval), while largely omitting quantitative data for traditional distortion-based attacks .

---

> ### Author Rebuttal · Authors · 2026-03-29
>
> We thank the reviewer for the thoughtful and constructive feedback, and for recognizing DAWN's motivation and efficiency.
>
> **(1) Distortion-based baselines:**
> We agree this strengthens the evaluation and have now completed full experiments across six distortion attacks spanning geometric, compression, smoothing, additive, and photometric perturbations, evaluated on all 10 watermarking schemes.
>
> Attack success rate alone is not a meaningful metric for evaluating distortion attacks. Our threat model (Section 4.1) requires simultaneous watermark removal AND perceptual fidelity. An attack that destroys the image to remove the watermark is not a practical alternative to DAWN. Both objectives must be satisfied together.
>
> Results across all 10 schemes consistently reveal two regimes, neither of which satisfies both objectives simultaneously:
>
> (i) High-success distortions fail on quality. Attacks that achieve meaningful removal do so only at severe perceptual cost – PSNR < 15 dB, LPIPS > 0.45, SSIM < 0.50. Images at this distortion level are immediately identifiable as manipulated and unusable for any realistic downstream purpose. The adversary has not achieved a practical attack.
>
> (ii) Quality-preserving distortions (JPEG, Blur) maintain good perceptual quality but fail entirely on removing semantic watermarks. Attack success is 0% on Tree-Ring, 0-24% on Zodiac – exactly the schemes where robust removal matters most and where DAWN provides its primary contribution. Higher the distortion intensity, higher is the attack success.
>
> The full per-metric results (PSNR, LPIPS, SSIM,  SSIM$\_{lum}$, CLIP, CLIP$\_{lum}$, Attack Succ.) across all
> 10 schemes and 6 distortion types will be provided in the camera-ready. The pattern is unambiguous: no distortion simultaneously achieves high success AND acceptable quality.  Dual-domain projection provides a capability that no simple distortion can replicate, because distortions can only trade one objective for the other while DAWN navigates both.
>
> **(2) Representative bias in qualitative figures:** We appreciate this observation. Quantitative evaluation already covers all 10 methods comprehensively in Table 2. For qualitative figures, Tree-Ring is used as the primary case study because it is the most
> challenging semantic watermark and best illustrates the structural-vs-chromatic preservation distinction that DAWN achieves. Figures 1 and 2 span different watermark types. We would include additional qualitative examples in the rebuttal, but are unable to do so due to the rebuttal format constraints. These will be included in the camera-ready.
>
> **(3) Conflict between quantitative metrics and perceived fidelity:**
>
> Figure 7 directly and empirically resolves this question. At T=2, pixel-only regeneration achieves higher PSNR/SSIM/CLIP and lower LPIPS than DAWN, yet fails to remove the watermark and produces heavily blurred images with melted textures and lost facial detail. Its PickScore (human preference metric) is substantially lower than DAWN's. This demonstrates concretely that standard metrics are biased toward low-frequency similarity and reward blurring as "quality," while PickScore correctly captures human-perceived fidelity. DAWN scores lower on pixel-level metrics precisely because it reconstructs high-frequency structural detail,  which PSNR/SSIM treat as "error" but humans recognize as content. This metric bias is analyzed in detail in Appendix A.4.
>
> **(4) Pipeline figure:**
> A comprehensive pipeline overview figure has been created, explicitly illustrating the two-stage attack core, the role of each module, and the decoupling of the optional perceptual alignment step from the attack mechanism. We are unable to share it within the rebuttal format, but it will be included in the camera-ready.
>
> If you found our responses satisfactory, we hope you will consider increasing the score further towards acceptance. We are happy to engage with more discussion should you have any questions.

---

> > ### Author Rebuttal · Reviewer_durv · 2026-04-06
> >
> > The authors have addressed my concerns in the rebuttal. I will raise my score to 5 (Accept).

---

> > > ### Author Response · Authors · 2026-04-06
> > >
> > > We thank the reviewer for the thoughtful feedback and for recognizing the motivation, efficiency, and contributions of our work.
> > >
> > > We especially appreciate the constructive suggestions on evaluation and presentation, which have helped us further strengthen the paper.
> > >
> > > We are grateful for the engagement during the discussion and for the positive reassessment.

---

### Official Review · Reviewer_doaq · 2026-03-18

**Soundness:** 3
**Presentation:** 3
**Significance:** 3
**Originality:** 3
**Overall Recommendation:** 4
**Confidence:** 1

**Summary:**

The paper focuses on the problem of watermark removal. It proposes a training-free, forward-pass-based attack method that can achieve a high attack success rate. The attack mainly relies on signal reconstruction or refinement in both the frequency domain and the semantic domain.

**Compliance With Llm Reviewing Policy:**

Affirmed.

**Final Justification:**

The rebuttal has addressed my concerns well.

**Key Questions For Authors:**

I am not familiar with this field, so I only have one main question. I think there is still significant room for improvement in the objective evaluation. At present, it is difficult for me to judge the novelty of the proposed method, because I do not have a clear understanding of how strong the baseline methods are and how meaningful the reported improvements are relative to them.

**Strengths And Weaknesses:**

1. I am not very familiar with this field, and my experience with watermark attack and defense methods is limited. Therefore, my comments may be relatively general.
2. I notice that some of the successfully attacked images exhibit a rather distinctive and somewhat unnatural visual style. In particular, the color composition appears slightly inconsistent, and many bright colors seem to be overly preserved or amplified. Is this a limitation of the proposed method?
3. In Table 2, I noticed that the evaluation metrics include {XXX}_{lum}. Why are those metrics chosen? Does it accurately reflect the (distortion/perceptual) quality of the attacked images? I am not sure whether a pretrained CLIP-based metric is appropriate for evaluating luminance-only image quality, because CLIP is not specifically designed to assess this type of visual quality.

---

> ### Author Rebuttal · Authors · 2026-03-28
>
> We thank the reviewer for their thoughtful feedback. We address each concern below:
>
> **(1) Visual artifacts / color inconsistency:** The chromatic shifts are an inherent consequence of frequency-domain watermark removal, not an implementation artifact. Because watermarks like TREE-RING embed signals in frequency bands that also encode color, suppressing watermark energy inevitably perturbs chrominance. However, structural/semantic content remains intact: SSIM$\_{lum}$ $\approx$ 0.99 and CLIP$\_{lum} \approx$ 0.99 (Table 2) confirm preserved structure and semantics. We explicitly address this via perceptual alignment (Step 3), which restores natural colors without reintroducing watermarks; attack success remains 71-77% after alignment (Table 3, Figure 5). This represents a fundamental tradeoff in the removability-distortion-compute space (Figure 4), which we document in Section 6.3 and Appendix A.1.
>
> **(2) Luminance metrics (SSIM$\_{lum}$, CLIP$\_{lum}$):** We introduced these specifically to address a known limitation of standard metrics: PSNR, SSIM, and CLIP all conflate chromatic deviation with structural damage and cannot distinguish between the two. Since DAWN primarily perturbs chrominance rather than structure, evaluating on the luminance (Y) channel of YCbCr space isolates the structurally and semantically meaningful components humans use for content recognition.
>
> Regarding CLIP$\_{lum}$ specifically: we use it as a semantic similarity proxy on the Y channel rather than as an absolute quality score. The interpretive value lies in the pattern: when SSIM$\_{lum} \approx$ 0.99 AND CLIP$\_{lum} \approx$ 0.99 hold simultaneously alongside lower standard metrics (PSNR ~14 dB, SSIM ~0.45), this demonstrates that degradation is chromatic rather than structural or semantic, a distinction no single metric alone can make. We agree that dedicated luminance-quality metrics would be preferable and note this as a direction for future metric development.
>
> **(3) Significance of improvements and novelty:** The core novelty of DAWN is not a marginal improvement over existing methods but demonstrating that a previously infeasible attack regime is feasible.
>
> To clarify why the reported numbers are significant:
> Prior to DAWN, no single-image low-compute method could remove frequency-embedded semantic watermarks at all – the natural pixel-space baseline achieves 0% success on TREE-RING (Section 3, Figure 3). DAWN achieves 70-77% success on the same scheme with no optimization, no surrogate models, and a single forward pass. The only prior method achieving comparable success (imprint-removal, Müller et al., CVPR 2025) does so using iterative gradient-based optimization at a substantially higher compute cost. The significance is therefore not a question of degree but of kind: DAWN opens a new point in the removability–distortion–compute design space that did not exist before. Low-compute AND high-success, which prior work implicitly assumed was unachievable without expensive optimization.
>
> If you found our responses satisfactory, we hope you will consider increasing the score further towards acceptance. We are happy to engage with more discussion should you have any questions.

---

> > ### Author Rebuttal · Reviewer_doaq · 2026-04-04
> >
> > Thank you for the clarifications. The rebuttal has addressed my concerns. However, as I am not very familiar with this field, I will maintain my original score.

---

> > > ### Author Response · Authors · 2026-04-06
> > >
> > > We thank the reviewer for the thoughtful feedback and engagement with the paper.
> > >
> > > We appreciate the questions raised regarding evaluation and perceptual quality, which helped us clarify important aspects of our work.
> > >
> > > The feedback has been valuable in improving the presentation and interpretability of the results.

---

### Decision · Program_Chairs · 2026-04-30

**Decision:**

Accept (regular)

**Comment:**

The main concerns raised relate to presentation clarity, completeness of experimental comparisons, and interpretation of perceptual quality metrics. Some reviewers also questioned whether the perceptual alignment step is fully decoupled from attack success, and noted inconsistencies in describing the pipeline. After rebuttal, these issues were largely clarified. The authors provided more detailed explanations of the pipeline, expanded comparisons with additional baselines, and clarified the role of perceptual alignment as well as the evaluation protocol. Importantly, multiple reviewers explicitly acknowledged that their concerns were addressed and updated their assessments accordingly.

While some limitations remain, such as the inherent trade-off between removal strength and visual fidelity and dependence on detectable spectral artifacts, these are clearly discussed and reflect broader challenges in the problem setting rather than flaws of the method itself.

Overall, the paper offers a useful perspective on low-compute watermark removal and demonstrates convincing empirical results in this regime. Based on the positive reviewer consensus and satisfactory rebuttal, I recommend acceptance.